# Early and Long-Term Ototoxicity Noted in Children Due to Platinum Compounds: Prevalence and Risk Factors

**DOI:** 10.3390/biomedicines11020261

**Published:** 2023-01-18

**Authors:** Alberto Romano, Serena Rivetti, Francesca Brigato, Stefano Mastrangelo, Giorgio Attinà, Palma Maurizi, Jacopo Galli, Anna Rita Fetoni, Antonio Ruggiero

**Affiliations:** 1Pediatric Oncology Unit, Fondazione Policlinico Universitario Agostino Gemelli IRCCS, 00168 Rome, Italy; 2Otorhinolaryngology Unit, Head and Neck Department, Fondazione Policlinico Universitario Agostino Gemelli IRCCS, 00168 Rome, Italy; 3Unit of Audiology, Department of Neuroscience, University of Naples Federico II, Reproductive Sciences and Dentistry, 80133 Naples, Italy

**Keywords:** ototoxicity, childhood cancer survivors, cisplatin, carboplatin, cancer, children

## Abstract

Background: Platinum compounds are a group of fundamental chemotherapeutics used in the treatment of solid tumors, but they are burdened by side effects, such as ototoxicity. The objective of this study was to evaluate the incidence of ototoxicity caused by platinum compounds and the risk factors affecting its appearance/progression. Methods: Data from 53 patients who received platinum compounds and who had been off therapy for at least 5 years were analyzed. We collected data relating to audiometry conducted annually from the end of treatment and for at least 5 subsequent years, as well as information concerning the oncological history and comorbidities. Results: At the end of the treatment, 17 patients (32.08%) presented ototoxicity, according to the Boston SIOP Ototoxicity Scale; the risk factors included a higher serum creatinine value at diagnosis, having undergone cranial radiotherapy, and needing magnesium supplementation. After 5 years from the end of the treatment, the number of patients with exhibiting ototoxicity was 31 (58.5%); the factors that influenced the onset/progression of the damage were having undergone radiotherapy (HR 1.23; *p* < 0.01) and having received therapy with aminoglycosides (HR 1.27; *p* < 0.01). Conclusions: Ototoxicity caused by platinum compounds can occur even after the conclusion of the treatments, and the factors affecting its progression are radiotherapy and the aminoglycosides therapy.

## 1. Introduction

In recent decades, the use of international cooperative protocols of treatment for pediatric cancer patients has resulted in a marked improvement in their survival [1]. The improvement in the survival of childhood cancer patients made it possible to observe the long-term side effects of cancer treatments on various organs such as the kidneys, heart, musculoskeletal system, nervous system, and gastrointestinal system [2,3,4,5,6,7]. 

Platinum compounds are among the pivotal drugs used in the treatment of pediatric cancer patients [8]. They are used in the treatment of numerous cancers, including germ cell tumors, hepatoblastoma, medulloblastoma, brain tumors, neuroblastoma, osteosarcoma, and refractory lymphomas [9]. However, their use is burdened by short- and long-term side effects, such as hearing damage [10]. Ototoxicity from platinum compounds is due to the degeneration of the hair cells of the ear caused by oxidative stress [11], inflammation, and the activation of the p53-dependent signaling pathways [12]. Hearing loss from platinum compounds is bilateral, sensorineural, and permanent [13] and can occur during the course of the treatment, at the end of the treatment, or occasionally, it can be perpetuated even after the suspension of therapy [14]. 

Numerous factors can affect the appearance of ototoxicity, including genetic predisposition, the primary site of the tumor, concomitant radiotherapy, exposure to other ototoxic drugs, the presence of kidney damage, and electrolyte disturbances [10]. Although ototoxicity does not represent a life-threatening complication, it does have significant effects on a patient’s quality of life, often compromising the cognitive development of the child and impairing social skills [15]. The early identification of hearing damage and the monitoring of its risk factors could allow for the implementation of measures that improve the outcome of cancer survivors [16,17].

In this study, we analyzed the prevalence of early- and late-onset ototoxicity and the risk factors for its onset/progression in pediatric patients undergoing treatment with platinum compounds.

## 2. Materials and Methods

### 2.1. Study Endpoints

The primary endpoint was:to describe the prevalence of early- and late-onset ototoxicity in pediatric cancer survivors treated with platinum compounds.

The secondary endpoints were:to evaluate the severity of ototoxicity according to the SIOP Boston Ototoxicity Scale;to analyze the clinical characteristics of patients with early- and late-onset hearing loss from platinum compounds;to analyze the potential risk factors for progression of hearing damage from platinum compounds after the end of treatment.

### 2.2. Study Design, Patient Characteristics, Inclusion and Exclusion Criteria

In this observational study, data from 90 pediatric patients treated with cisplatin and/or carboplatin, consecutively admitted at the Pediatric Oncology Unit of the Fondazione Policlinico Universitario Agostino Gemelli IRCCS from January 2007 to December 2017, were retrospectively analyzed. The audiological evaluation was performed in the Service of Audiology of the Fondazione Policlinico Universitario Agostino Gemelli IRCCS. 

Informed consent was obtained from the parents or the legal guardians of the enrolled patients. The study was carried out fin accordance with the Declaration of Helsinki, and it was approved by our ethical committee (protocol ID: 0019971/20-ID 3074). 

Inclusion criteria were: cancer diagnosis at age <18 years;having undergone therapy with cisplatin and/or carboplatin;age at the first platinum course ≤18 years;availability of audiological baseline evaluation before the beginning of treatment, audiological re-evaluation prior to each chemotherapy cycle, and a yearly audiological long-term follow-up for at least 5 years after the end of chemotherapy.

Exclusion criteria were:age at first platinum course >18 years;diagnosis of hearing loss at the baseline evaluation before chemotherapy;having received the last administration of platinum compounds less than 5 years previously;absence of audiological evaluation at one or more time-points.

### 2.3. Clinical Data 

For each patient, the following data were collected:demographic characteristics (age and sex);data related to the neoplastic disease (histology, primarily affected site, date of diagnosis, presence of metastasis at diagnosis);type and duration of the treatment (type and dose of platinum compounds, site and dose of radiotherapy, neurosurgery);data related to comorbidity (serum creatinine at diagnosis; creatinine clearance at diagnosis; having required magnesium supplementation during treatment with platinum compounds; having received diuretics during treatment with platinum compounds; having received aminoglycosides, and dose during treatment with platinum compounds).

### 2.4. Audiological Assessment

Audiological assessments were performed by the same team of doctors and audiologists. All patients underwent at least an annual evaluation. In patients with hearing loss, the evaluation was carried out every six months. All audiological assessments included otoscopy and tympanometry, as well as acoustic reflex measurements in order to exclude middle ear pathology (Grason Stadler, Eden Prairie, MN, USA). The pure tone air conduction thresholds were evaluated at frequencies of 0.25, 0.5, 1, 2, 4, and 8 kHz in hearing at the decibels level (dB HL). Pure tone bone conduction thresholds were evaluated at frequencies of 0.25, 0.5, 1, 2, and 4 kHz to determine the type of hearing damage (i.e., conductive, sensorineural, or mixed). Visual reinforcement audiometry was performed in a calibrated sound field or via earphones for children from 6 to 30 months of age. In a sound field, the results reflect hearing in the best ear, if there is a difference in hearing between the ears. In general, frequencies at 0.5, 1, 2, and 4 kHz were obtained when visual reinforcement audiometry was used to measure hearing. Conditional play audiometry (train show) was used for children to be able to cooperate (from 30 months to 5 years). Standard pure tone audiometry was used for older children (>5 years) under standard conditions using an Amplaid 319 audiometer (Amplaid Inc., Milan, Italy) in a double-walled and soundproofed room for conventional testing frequency ranges from 0.25 to 8 kHz (see www.asha.organization/politics, accessed on 17 December 2022). The audiological recordings were reviewed and classified according to the SIOP (International Society of Pediatric Oncology) Boston Ototoxicity Scale (grade 0–4). Consequently, with our protocol [18], when a child was diagnosed with SIOP grade >2 ototoxicity, administration of cisplatin was switched to carboplatin. Progressive hearing loss was been reported in children who presented any SIOP grade >0 at the end of treatment and that progressed in the subsequent follow-ups, while the children who developed an SIOP grade >0 after the end of treatment were referred to as having late onset hearing loss.

### 2.5. Statistical Analyses

Continuous data are given as median and standard deviation (SD) and were compared using Mann–Whitney U tests. Categorical data are presented as absolute and percentage frequencies and were compared using the Chi-squared test or the Fisher’s exact test, where appropriate. 

The time of appearance of the late onset hearing loss was defined as the months elapsed between the end of therapy and the appearance of the damage. The time of progression of the progressive hearing loss was defined as the months elapsed between the end of therapy and the appearance of progression of the damage. The prognostic significance of the covariates was assessed with univariate and multivariate analyses using a Cox proportional hazards model, expressed as hazard ratios (HR) and 95% confidence intervals (95% CI). All variables significant in univariate analyses were entered into multivariate models. In univariate and multivariate analyses, we considered the time of appearance of late onset hearing loss and the time of progression of progressive hearing loss as a single group in order to be able to analyze the risk factors that determine the perpetuation of hearing damage caused by platinum compounds after the end of treatment. All statistical analyses were performed using XLSTAT version 2021.3.1 by Addinsoft; *p* values < 0.05 were considered statistically significant. 

## 3. Results

### 3.1. Study Population, Demographics, and Clinical Characteristics

Among the 90 patients treated with cisplatin and/or carboplatin at our center between January 2007 to December 2017, 53 were included in the study. A total of 26 patients were excluded due to the absence of audiological evaluation in one or more time-points (most of whom did not adhere to all follow up checks); 11 patients were excluded because they received the last administration of platinum compounds more than 5 years previously.

The demographic and clinical characteristics of the enrolled population are reported in Table 1.

A total of 30 (56.6%) patients were male and 23 (43.4%) were female, with a median age of 6.56 (SD 4.5) years old. A total of 44 (83%) patients had brain tumors, and 9 (16.9%) had a primitive extracranial localization. A total of 41 of the 44 patients with brain tumors underwent a neurosurgery intervention, and 21 received cerebral radiotherapy. Carboplatin was administered to 44 (83%) patients (cumulative dose of 3219.6 [SD 2476.2]), while cisplatin was administered to 29 (54.7%) patients (cumulative dose of 367 [SD 218.9]). The number of patients who received both platinum compounds was 18 (33.9%). During treatment with platinum compounds, hypomagnesaemia was found in 8 (15.1%) patients, for which intravenous supplementation was administered. A total of 23 (43.4%) patients underwent aminoglycoside therapy due to febrile neutropenia during treatment with platinum compounds. A diuretic was used in only one patient. No patient presented anomalies in renal function at the start of treatment, and in no case was there any evidence of alteration in renal function during treatment with platinum compounds or during follow-up.

### 3.2. Audiological Data, Prevalence of Ototoxicity, and Its Severity

At the end of treatment, 17 (32.1%) patients had developed ototoxicity: 5 (29.4%) had class 1 ototoxicity according to the SIOP Boston Ototoxicity Scale, 3 (17.7%) had class 2 ototoxicity, 1 (5.9%) had class 3 ototoxicity, and 8 (47%) had class 4 ototoxicity. In 6 patients, it was necessary to replace cisplatin with carboplatin during treatment due to the appearance of SIOP grade >2 ototoxicity (1 patient with class 2 ototoxicity at the end of treatment and 5 patients with class 4 ototoxicity at the end of treatment).

Five years after the end of treatment, 31 (58.5%) patients exhibited hearing damage; 8 (25.8%) had class 1 ototoxicity, 2 (6.5%) had class 2 ototoxicity, 1 (3.2%) had class 3 ototoxicity, and 20 (64.5%) had class 4 ototoxicity, according to the SIOP Boston Ototoxicity Scale. Thus, 5 years after the end of treatment, only 22 patients had normal hearing.

Figure 1 shows the prevalence of ototoxicity and its severity at the end of treatment and 5 years after the end of treatment.

### 3.3. Characteristics of the Population with Hearing Damage from Platinum Compounds at the End of Treatment

At the end of the treatment, 17 patients (32.1%) showed ototoxicity, while 36 patients (67.9%) had normal hearing. No statistically significant differences were observed between patients with ototoxicity and patients with normal hearing at the end of treatment in relation to the age at diagnosis, the age at the beginning of treatment and at the end of treatment, nor to the dose of cisplatin and/or carboplatin.

A statistically significant difference was instead observed in relation to the creatinine value at the beginning of the treatment. In fact, patients who showed ototoxicity at the end of treatment had, on average, higher creatinine values (0.58; SD 0.24) before starting chemotherapy than patients with normal hearing (0.42; SD 0.17) (p 0.04).

The use of other ototoxic drugs, such as aminoglycosides, does not seem to have affected the onset of ototoxicity at the end of treatment in our group of patients, while the presence of alterations in the blood concentration of magnesium and the consequent need for its intravenous integration influenced the onset of hearing impairment. In fact, of the 17 patients with ototoxicity, 29.4% needed magnesium supplementation, while in patients with normal hearing only 8.3% needed magnesium (p 0.04).

Most of the patients with ototoxicity at the end of treatment had undergone cranial radiotherapy (76.5%). Of the 21 patients who underwent radiotherapy, 13 exhibited ototoxicity and only 8 did not. On the other hand, the dose of radiotherapy administered does not seem to influence the onset of ototoxicity.

Table 2 compares the presence of risk factors for the onset of ototoxicity between the group of patients with ototoxicity and patients with normal hearing at the end of treatment.

### 3.4. Characteristics of the Population with Late-Onset Hearing Loss/Progressive Hearing Loss

A total of 5 years after treatment, the number of patients with hearing loss was 31 (58.5%), while the number of patients with normal hearing was 22 (41.5%). Of the 31 patients with hearing impairment 5 years after the end of treatment, 14 showed the condition of late onset hearing loss. Progressive hearing loss was observed in 7 patients who presented with ototoxicity at the end of treatment. Therefore, 21 (39.6%) patients showed a worsening of hearing abilities in the 5 years following the end of the treatment, 14 passed from class 0 to a class >1 on the SIOP Boston Ototoxicity Scale, (5 [35.7%] had class 1 ototoxicity, 1 [7.15%] had class 2 ototoxicity, 1 [7.15%] had class 3 ototoxicity, and 7 [50%] had class 4 ototoxicity according to SIOP Boston Ototoxicity Scale), and 7 presented a progression of the damage which was already present at the end of the treatment (3 patients passed from class 1 to class 4, 2 patients passed from class 2 to class 4, 1 patient from class 2 to class 3, and 1 patient from class 3 to class 4) (see Figure 2).

We compared the characteristics of the patients who experienced a worsening of hearing condition over the 5-year follow-up period with those who maintained stable hearing, and we observed that no statistically significant differences were present between the two groups in relation to the age at diagnosis, the age at the beginning of treatment and the end of treatment, or the dose of cisplatin and/or carboplatin.

In contrast to what was observed at the end of the treatment, the average value of serum creatinine before the beginning of the treatment and the need for Mg intravenous integration were not found to be different between the two groups of patients. However, the percentage of patients who received aminoglycosides during chemotherapy was higher among patients with a progression of the damage (61.9%) than among those with stable hearing (37.5%) (*p* = 0.02).

Five years after the end of the therapy, the number of patients with late-onset hearing loss/progressive hearing loss who had undergone radiotherapy was 13, statistically different from those with stable hearing (*p* = 0.007).

Table 3 compares the characteristics of the patients of the group with late-onset hearing loss/progressive hearing loss and patients with stable hearing at the end of the 5 years of observation.

### 3.5. Risk Factors for Progression of Acoustic Damage in Patients with Late-Onset Hearing Loss/Progressive Hearing Loss

The median time to progression of hearing damage was 25.4 months (SD 16.6). Figure 3 shows the progression of acoustic damage in the population over the 5 years following the end of treatment.

As shown in Table 4, the factors that influenced the progression of the damage in the 5 years following the end of the treatment were having undergone radiotherapy (HR 1.23; *p* < 0.01) and having received aminoglycosides (1.27; *p* < 0.01).

## 4. Discussion

The prevalence of early-onset hearing loss, which was defined as ototoxicity registered at the end of treatment, in our sample of patients was 32.1%. This prevalence is lower than those observed in previous studies [19], and this result was probably linked to the composition of the group of patients we studied. In fact, a higher prevalence of ototoxicity was observed for doses of cisplatin greater than 400 mg/m^2^, while in our population, the mean dose of cisplatin was 367 mg/m^2^. Furthermore, 54.7% of our patients received cisplatin during treatment, while 83% received carboplatin; previous studies have shown that cisplatin is burdened by greater ototoxicity than carboplatin [20]. The lower prevalence of hearing damage in our population is therefore attributable to these two factors.

As evidenced by previous studies [19], a lower age at diagnosis could represent a risk factor for the appearance of hearing damage. Consequently, in our sample, we expected to observe a lower mean age at the start of treatment in patients with early-onset hearing loss than in those with normal hearing, but we did not observe a statistically significant difference between the two groups. Instead, we observed a difference in the plasma creatine value at the beginning of treatment, with greater value in patients with early-onset hearing loss. The platinum compounds are mainly eliminated by the kidney [21] and therefore, decreased renal function can cause an inferior elimination of the drug, with a consequently greater hearing damage. Furthermore, platinum compounds act directly on the kidneys, being able to alter their functions and the tubular reabsorption of electrolytes, in particular of magnesium, with a consequent reduction in the magnesium content in the endolymph and perilymph of the ear. This mechanism makes it easier for platinum compounds to damage the inner ear [22,23,24,25]. As proof of this, in the enrolled population, we observed that patients with early-onset hearing loss needed more magnesium supplementation. However, we did not observe a difference in co-treatment with aminoglycosides between patients with early-onset hearing loss and those with normal hearing. This element suggests the need for the careful monitoring and supplementation of magnesium during platinum compound therapy.

As previously noted, even in our study population, most patients with ototoxicity had undergone cranial radiotherapy. In fact, this is an important risk factor for the onset of hearing damage from platinum compounds [26].

In the population with early-onset hearing loss, we also observed a higher percentage of patients undergoing neurosurgical surgery than in the population with normal hearing at the end of treatment. This was probably due to the primary location of the tumor and damage to the nerve structures necessary to achieve tumor removal.

Five years after the end of treatment, the prevalence of ototoxicity increased to 58.5%. This data demonstrates that the damage caused by platinum compounds can occur and worsen, even after the end of the treatment. In fact, 14 patients who did not have hearing loss at the end of treatment experienced late-onset hearing loss in the following 5 years, and 7 patients experienced a worsening of their conditions. In analyzing the characteristics of these 21 patients, which made up 39.6% of the population, we observed that having received aminoglycosides and having undergone cranial radiotherapy were the only distinguishing features with respect to patients with stable hearing. These two factors have also demonstrated statistical significance in the risk analysis, further highlighting their ability to facilitate the continuation over time of hearing damage from platinum compounds.

As previously supposed, the damage to the outer hair cells by aminoglycosides, which are the major target of cisplatin ototoxicity [27,28], causes an effect on hearing over a longer period than do the platinum compounds, justifying their influence on the occurrence of late-onset hearing loss and progressive hearing loss.

Radiotherapy, on the other hand, in addition to facilitating the early-onset of damage from platinum compounds, has shown to be a risk factor for the worsening of hearing loss over time, even after the end of treatment [15].

On the other hand, unlike early-onset hearing loss, in patients with late-onset hearing loss/progressive hearing loss, we did not observe differences in relation to the value of plasma creatinine or in the need for magnesium supplementation. These two elements are probably implicated in the early hearing damage from platinum compounds, but do not act in favoring the continuation of the damage after the end of the treatment.

These data demonstrate that the damage from platinum compounds is perpetuated over time and can manifest itself/progress even after the conclusion of the treatment. Therefore, surveillance cannot end with the conclusion of the treatment, but must continue over time to allow for the early identification of the onset of hearing damage. Furthermore, for patients undergoing cranial radiotherapy and aminoglycoside therapy, monitoring must be even more accurate, since these patients are at risk for the progression of ototoxicity after the end of treatment. In our study, we evaluated a 5-year follow-up; however, especially in patients undergoing cranial radiotherapy and those who receive aminoglycosides, the follow-up should be continued even beyond 5 years, given the possibility of late ototoxicity and the worsening of the damage auditory system.

## 5. Conclusions

The prevalence of early ototoxicity in our population was 32.1%, and the factors influencing its appearance included having undergone radiotherapy, having had a higher serum creatinine value at the beginning of treatment, and having needed magnesium supplementation.

A total of 5 years after the end of the treatment, the prevalence of ototoxicity increased up to 58.5%, and 39.6%. of the patients experienced a worsening of hearing in the 5 years after the end of treatment. The factors that influenced the worsening of the damage in the 5 years after the conclusion of the therapy were radiotherapy and aminoglycoside therapy.

Therefore, the audiological follow-up of childhood cancer survivors cannot be concluded at the end of the treatment, but must continue over time, with particular attention paid to subjects undergoing radiotherapy and aminoglycoside therapy who could undergo a worsening condition and require medical interventions aimed at the recovery of lost function before further progression.

## Figures and Tables

**Figure 1 biomedicines-11-00261-f001:**
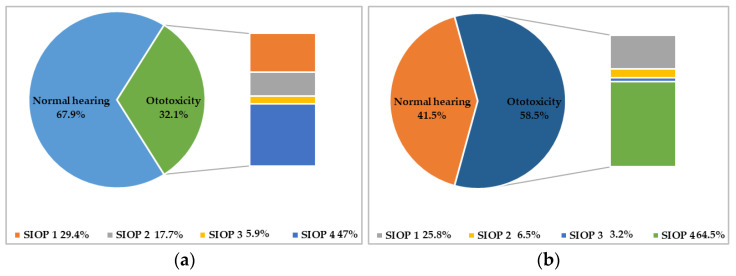
Prevalence and severity of ototoxicity according to the SIOP Boston Ototoxicity Scale. In (**a**) is reported the prevalence and severity of ototoxicity at the end of treatment; in (**b**) is reported the prevalence and severity of ototoxicity 5 years after the end of treatment.

**Figure 2 biomedicines-11-00261-f002:**
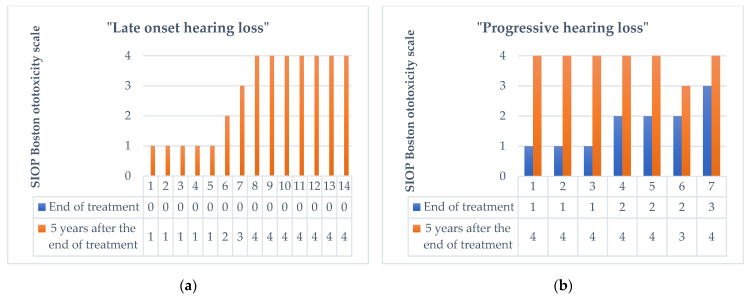
Change in the degree of ototoxicity according to the SIOP Boston Ototoxicity Scale 5 years after the end of therapy in 21 patients with late onset hearing loss (**a**), and progressive hearing loss (**b**).

**Figure 3 biomedicines-11-00261-f003:**
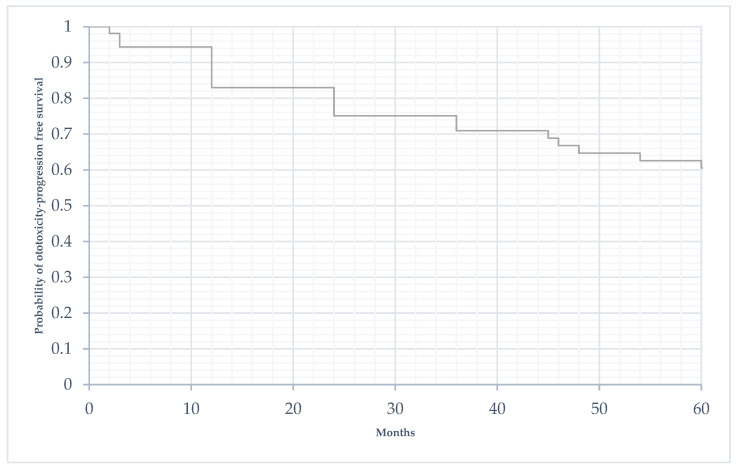
Time to progression of acoustic damage in the 5 years following the end of treatment (in months).

**Table 1 biomedicines-11-00261-t001:** Demographic and clinical characteristics of the patients enrolled (*n* = 53).

Characteristics	Number of Patients (%) or Median [SD]
**Age at diagnosis, in years**	6.56 [4.5]
**Male sex**	30 (56.6)
**Histology**	
Low Grade Glioma	20 (37.7)
Medulloblastoma	13 (24.5)
Wilms Tumor	2 (3.8)
Germ cells tumor	6 (11.3)
Retinoblastoma	6 (11.3)
Soft tissue sarcomas	1 (1.9)
Neuroblastoma	3 (5.7)
Carcinoma	2 (83.8)
**Primary localization**	
Cerebral	44 (83)
Extra-cerebral	9 (16.9)
**Presence of metastases at diagnosis**	10 (18.9)
**Neurosurgery**	41 (77.4)
**Cerebral radiotherapy**	21 (39.6)
**Radiotherapy dose in Gy**	35.6 [13.9]
**Carboplatin**	44 (83)
**Carboplatin dose in mg/m^2^**	3219.6 [2476.2]
**Cisplatin**	29 (54.7)
**Cisplatin dose in mg/m^2^**	367 [218.9]
**Carboplatin and cisplatin**	18 (33.9)
**Age at the start of treatment, in years**	7.3 [5.2]
**Age at the end of treatment, in years**	8.22 [5.2]
**Mg supplementation**	8 (15.1)
**Therapy with diuretics**	1 (1.9)
**Aminoglycoside therapy**	23 (43.4)
**Aminoglycoside cumulative dose, in mg**	6104.2 [9391.5]
**Serum creatinine, in mg/dL ***	0.37 [0.2]
**Creatinina clearance, in mL/min ***	157.3 [8.55]

* Before the start of treatment.

**Table 2 biomedicines-11-00261-t002:** Comparison between patients with hearing impairment and patients with normal hearing at the end of treatment.

Ototoxicity at The End of Treatment	Yes	No	*p* Value
Number of patients (%)	17 (32.1%)	36 (67.9%)	<0.01
Age at diagnosis, in years (SD)	8.9 (6.7)	5.4 (4.3)	0.08
Age at the start of treatment, in years (SD)	8.09 (6)	6.5 (4.69)	0.16
Age at the end of treatment, in years (SD)	10.01 (5.87)	7.38 (4.71)	0.1
Duration of treatment, in years (SD)	1.04 (0.74)	0.90 (0.84)	0.64
N pts who received carboplatin (%)	13 (76.5%)	31 (86.1%)	0.38
Cumulative dose of carboplatin, in mg/m^2^ (SD)	3679.16 (2960.23)	3041.77 (2292.27)	0.61
N pts who received cisplatin (%)	12 (70.6%)	17 (47.2%)	0.11
Cumulative dose of cisplatin, in mg/m^2^ (SD)	417.9 (236.5)	326.3 (202.7)	0.25
N pts who received carboplatin and cisplatin (%)	6 (35.3%)	12 (33.3%)	0.78
Pts who received aminoglycosides (%)	10 (58.8%)	13 (36.1%)	0.12
Cumulative dose of aminoglycosides, in mg (SD)	7286 (10813.7)	5259.9 (8558.6)	0.45
Creatinine at the beginning of treatment, in mg/dL (SD)	0.58 (0.24)	0.42 (0.17)	0.04
Creatinine clearance at the beginning of treatment (SD)	154.56 (57.6)	158.87 (55.9)	0.88
N pts who received cranial radiotherapy (%)	13 (76.5%)	8 (22.3%)	<0.01
Radiotherapy dose, in Gy (SD)	33.95 (14.4)	38.88 (13.08)	0.57
N pts who needed Mg supplementation (%)	5 (29.4%)	3 (8.3%)	0.04
N pts subjected to neurosurgical intervention (%)	16 (94.1%)	25 (69.4%)	0.04

N: number; pts: patients; Mg: magnesium.

**Table 3 biomedicines-11-00261-t003:** Comparison between patients with late-onset hearing loss/progressive hearing loss and patients with stable hearing at the end of the 5 years of observation.

Patients with Late-Onset Hearing Loss/Progressive Hearing Loss	Yes	No	*p* Value
Number of patients (%)	21 (39.6%)	32 (60.4%)	<0.01
Age at diagnosis, in years (SD)	8.15 (6.19)	5.52 (4.67)	0.08
Age at the start of treatment, in years (SD)	8.27 (5.59)	6.62 (4.96)	0.21
Age at the end of treatment, in years (SD)	9.11 (5.45)	7.64 (5.03)	0.31
Duration of treatment, in years (SD)	0.83 (0.68)	1.01 (0.87)	0.38
N pts who received carboplatin (%)	17 (80.9%)	27 (84.4%)	0.74
Cumulative dose of carboplatin, in mg/m^2^ (SD)	3134.38 (2532.3)	3279.18 (2489.7)	0.78
N pts who received cisplatin (%)	14 (66.7%)	15 (46.9%)	0.16
Cumulative dose of cisplatin, in mg/m^2^ (SD)	360.42 (164.59)	372.33 (260.16)	0.69
N pts who received carboplatin and cisplatin (%)	9 (42.9%)	9 (28.1%)	0.42
Pts who received aminoglycosides (%)	13 (61.9%)	12 (37.5%)	0.02
Cumulative dose of aminoglycosides, in mg (SD)	3603.07 (1843.69)	9059.9 (13455.17)	0.92
Creatinine at the beginning of treatment, in mg/dL (SD)	0.46 (0.21)	0.48 (0.21)	0.75
Creatinine clearance at the beginning of treatment (SD)	135.12 (44.29)	174.31 (58.59)	0.06
N pts who received cranial radiotherapy (%)	13 (61.9%)	8 (25%)	0.007

* N: number; pts: patients; Mg: magnesium.

**Table 4 biomedicines-11-00261-t004:** Univariate and multivariate analysis of the onset/progression of acoustic damage in the 5 years following the conclusion of treatment.

	Univariate Analysis		Multivariate Analysis	
	HR (95%CI).	*p* Value	HR (95%CI)	*p* Value
Underwent neurosurgery	0.68 (0.15–1.72)	0.27		
Underwent cranial radiotherapy	1.14 (0.38–1.9).	0.01.	1.23 (0.51–1.95)	<0.01
Received carboplatin	0.23 (0.23–2.72)	0.71		
Received cisplatin	0.59 (0.22–1.38)	0.2		
Received carboplatin and cisplatin	0.17 (0.01–0.91)	0.99		
Required magnesium supplementation	0.39 (0.23–2.02)	0.47.		
Received aminoglycosides	1.19 (0.42–1.96)	0.01	1.27 (0.56–1.99)	<0.01
Age at diagnosis	0.07 (0.99–1.15)	0.05		
Radiotherapy dose	0.03 (0.98–1.06)	0.2		
Age at the beginning of treatment	0.05 (0.98–1.15)	0.11		
Duration of chemotherapeutic treatment	0.17 (0.48–1.46)	0.5		
Serum creatinine at diagnosis	0.29 (0.06–9.02)	0.8		

## Data Availability

The data presented in this study are available on request from the corresponding author. The data are not publicly available due to privacy.

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
