# Peer review of "Early and Long-Term Ototoxicity Noted in Children Due to Platinum Compounds: Prevalence and Risk Factors"

_biomedicines, 2023, doi:10.3390/biomedicines11020261_

Round 1

Reviewer 1 Report

 In this research, authors analysed the prevalence of early and late onset ototoxicity and the risk factors for its onset/progression in paediatric patients undergoing treatment with platinum compounds. The introduction provides a short and objective background to the subject, with a clear aim, explained better in the first subsection of the methods section. The experiment design was well described, however I did not clearly find information on which samples were controls and which were cases, and did not understand how the comparisons were performed in terms of group vs group. Results were very descriptive. It would be better to highlight the message you want to deliver from the result you are describing or at least highlight the results you want the read to be more aware of. Discussion and conclusion covered the relevant results bringing up the general message the paper intended to deliver. Therefore, I consider the publication of the manuscript after minor revisions, some of which I suggest below:

It would be better for the reader if the result subtitles showed the general result presented in the subsection. For example, in the subsection 3.2, it would be like this: “Hearing damage increases after 5 years treatment”.

In figure 1, the title of the chart on the left is wrong. It would be better to describe them as A and B, and put the title or the description in the caption.

Remove “the characteristics of” from the table 2 and table 3 titles. (Lines 208 and 245).

Replace “/” in the figure 2 caption by “and”. (Line 226).

Author Response

In this research, authors analysed the prevalence of early and late onset ototoxicity and the risk factors for its onset/progression in paediatric patients undergoing treatment with platinum compounds. The introduction provides a short and objective background to the subject, with a clear aim, explained better in the first subsection of the methods section. The experiment design was well described, however I did not clearly find information on which samples were controls and which were cases, and did not understand how the comparisons were performed in terms of group vs group. Results were very descriptive. It would be better to highlight the message you want to deliver from the result you are describing or at least highlight the results you want the read to be more aware of. Discussion and conclusion covered the relevant results bringing up the general message the paper intended to deliver.

Thanks reviewer 1 for the comments. We have tried to improve the article with your suggestions. Changes to the manuscript made in response to your comments are highlighted in yellow.

In paragraph 3.3. we have inserted the sentence “patients with ototoxicity and patients with normal hearing at the end of treatment” to better specify the groups between which the comparison was made. Similarly in paragraph 3.4. we have highlighted the sentence “patients who experienced a worsening of hearing condition over the 5-year follow-up with those who maintained stable hearing” to highlight which groups were compared. moreover, in the tables we have rewritten the statistically significant results in bold to underline their importance.

Therefore, I consider the publication of the manuscript after minor revisions, some of which I suggest below:

It would be better for the reader if the result subtitles showed the general result presented in the subsection. For example, in the subsection 3.2, it would be like this: “Hearing damage increases after 5 years treatment”.

In the titles of the paragraphs of the results we have preferred to describe the content of the paragraph and not the results contained in order to insert our comments only in the discussion. In this way we hope that the text is as descriptive as possible.

In figure 1, the title of the chart on the left is wrong. It would be better to describe them as A and B, and put the title or the description in the caption.

We modified the figure 1 as suggested.

Remove “the characteristics of” from the table 2 and table 3 titles. (Lines 208 and 245).

We changed the text as suggested.

Replace “/” in the figure 2 caption by “and”. (Line 226).

We changed the text as suggested.

Reviewer 2 Report

1. Check for the results of number of patients with ototoxicity as mentioned in abstract section, as it does not match with the obtained results.

2. Check for the sentence “Visual reinforcement audiometry was performed in a calibrated sound field o via earphones for children from 6 to 30 months of age. Sound field the results  reflect hearing in the best ear if there is a difference in hearing between the ears” and rewrite the same.

3. In Figure 1. label the figure as Figure 1a and 1b and check for the title of graph representing “ototoxicity scale at the end of treatment”.

4. Check for the font size in Figure 2.

5. In Figure 3, y-axis is not labelled in the graph.

6. “Disclaimer/Publisher’s Note” is included in the reference list. Check for the same.

Author Response

Thanks reviewer 1 for the comments. We have tried to improve the article with your suggestions. Changes to the manuscript made in response to your comments are highlighted in green.

  1. Check for the results of number of patients with ototoxicity as mentioned in abstract section, as it does not match with the obtained results.

We had made a mistake by inserting the number of patients with "late onset hearing damage"/"progressive hearing damage" instead of the total number of patients with ototoxicity 5 years after the end of treatment. We have corrected the error.

  1. Check for the sentence “Visual reinforcement audiometry was performed in a calibrated sound field o via earphones for children from 6 to 30 months of age. Sound field the results reflect hearing in the best ear if there is a difference in hearing between the ears” and rewrite the same.

We modified the sentence. Now it is “Visual reinforcement audiometry was performed in a calibrated sound field o via earphones for children from 6 to 30 months of age. In sound field the results reflect hearing in the best ear if there is a difference in hearing between the ears.”

  1. 3. In Figure 1. label the figure as Figure 1a and 1b and check for the title of graph representing “ototoxicity scale at the end of treatment”.

We changed Figure 1 as suggested.

  1. Check for the font size in Figure 2.

We modified the figure 2 as suggested.

  1. In Figure 3, y-axis is not labelled in the graph.

We modified the figure 3 as suggested.

  1. “Disclaimer/Publisher’s Note” is included in the reference list. Check for the same.

We modified it.
